# Fluroxypyr Inhibits Maize Growth by Disturbing the Diversity of the Endophytic Bacterial Communities in Maize Roots

**DOI:** 10.3390/microorganisms13040728

**Published:** 2025-03-24

**Authors:** Gangrui Zhang, Nan Liu, Shengbo Shi, Jinghua Li, Rui Geng, Longyu Fang, Yuanyuan Wang, Mingchun Lin, Junfeng Chen, Yanru Si, Kai Shan, Zeyun Zhou, Maoyu Men, Xiangren Qiao, Lujiang Hao

**Affiliations:** School of Bioengineering, Qilu University of Technology (Shandong Academy of Sciences), Jinan 250353, China; 10431221312@stu.qlu.edu.cn (G.Z.); 10431221234@stu.qlu.edu.cn (N.L.); 10431230730@stu.qlu.edu.cn (S.S.); 10431230807@stu.qlu.edu.cn (J.L.); 10431230774@stu.qlu.edu.cn (R.G.); 10431230825@stu.qlu.edu.cn (L.F.); 10431230832@stu.qlu.edu.cn (Y.W.); 10431240798@stu.qlu.edu.cn (M.L.); 10431240800@stu.qlu.edu.cn (J.C.); 10431240837@stu.qlu.edu.cn (Y.S.); kai_shan2017@163.com (K.S.); 202281010041@stu.qlu.edu.cn (Z.Z.); 202381013075@stu.qlu.edu.cn (M.M.); 202496063019@stu.qlu.edu.cn (X.Q.)

**Keywords:** fluroxypyr, herbicides, maize growth, bacterial community

## Abstract

Fluroxypyr (4-amino-3,5-dichloro-6-fluoro-2-pyridyloxyacetic acid) is a widely used herbicide sprayed on crops worldwide. The effects of fluroxypyr on maize growth and the soil microbial community structure have not been reported. In this study, the impacts of fluroxypyr on maize growth and the bacterial community structure in endophytes and rhizospheric/non-rhizospheric soils were evaluated. We found that the community structures of the non-rhizospheric and rhizospheric soils were similar. The alpha diversity showed that the richness of the endophytic communities in the mature maize roots was reduced after herbicide application. No statistically significant differences were observed between the fluroxypyr-treated and control soils in either the non-rhizospheric or rhizospheric soils. However, the composition of the endophytic bacterial community structure suggested that fluroxypyr led to a 59.1% reduction in the abundance of *Acinetobacter* and a 75.6% reduction in *Agrobacterium*, both of which are considered growth-promoting bacteria. In addition, we observed a negative effect of fluroxypyr on maize growth, including a decreased ear length and root size and a reduction in the 100-grain weight. In summary, our study suggests that fluroxypyr may negatively impact the mature growth of maize by reducing the abundance of *Bacillus kineticus* and *Agrobacterium tumefaciens* in the endophytic community of the mature root system.

## 1. Introduction

Maize is an important food crop that is relevant to agricultural production and economic development. Weeds and crops are herbaceous plants that compete for light, soil nutrients, and water resources, thus leading to crop reduction. Long-lasting symbiosis with crops may affect crop yields (Chauhan and Johnson 2011 [1]; Takim 2012 [2]). Many types of weeds are abundant in maize fields and produce damage. Weeds are a major obstacle to maize quality (Mehmeti et al., 2019 [3]).

Herbicides are widely used in agriculture and conserve time and labor while increasing efficiency and production. Mesarović et al. (2019) [4] found that the application of herbicides after emergence was the most important factor in reducing weed infestation. Previous studies have demonstrated that, although pesticides are applied to plants and soils to inhibit pests and weeds, they may also affect the soil properties, microbes, and hosts (Altman and Campbell 1977 [5]). Bhandari et al. (2024) [6] found that pesticides could affect soil enzyme activity. The results of Benito et al. (2021) [7] showed that, under good moisture and temperature conditions, the yield of Aflatoxin B1 in maize grains increased with the concentration of a glyphosate-based herbicide in line with the recommended field dose. Anderson et al. (1981) [8] found that the balance of fungi and bacteria in the soil treated with a high concentration of Verdasan was disrupted, and the microbial biomass decreased and could not be recovered for a long time.

The soil bacterial community is essential for soil quality, affects soil fertility, and directly influences plant growth and fitness (Gangola et al., 2022 [9]; Ramos et al., 2003 [10]). Herbicides may change the bacterial community structure and affect the microbial population size (Gangola et al., 2024 [6]), while influencing the endogenous bacteria in the plant root system (Kuklinsky-Sobral et al., 2004 [11]). Microbes are the main decomposers in ecosystems, and they serve as important ecosystem components (Naik et al., 2009 [12]; Qiu et al., 2012 [13]; Chen et al., 2013 [14], 2016 [15]). Plant endophytes and soil microorganisms have important effects on plant growth and soil pollutant bioremediation (Mahbub et al., 2016 [16]; Lin et al., 2016 [17]; Cheng et al., 2020 [18]; Ramakrishna et al., 2020 [19]). Plants influence the structure and composition of the endophyte and soil microbial communities (Esperschütz et al., 2009 [20]; Li et al., 2019 [21]). When confronted with external environmental stresses, plant roots secrete substances to cope with changes in the external environment that affect the composition and structure of microorganisms in the plant roots (Esperschütz et al., 2009 [20]). However, environmentally hazardous materials may have adverse effects on plants by affecting the plant root microbial community.

The impacts of herbicides on soil microbial communities include altering the microbial diversity and abundance, inhibiting the growth of certain beneficial microorganisms, and potentially promoting the proliferation of microbes with higher tolerance or stronger degradation capabilities (Yang et al., 2017 [22]; Pu et al., 2019 [23]). The long-term application of herbicides may lead to functional imbalances in microbial communities, reducing the stability of the soil ecosystem (Sanders and Pallett 1987 [24]). Fluroxypyr (4-amino-3,5-dichloro-6-fluoro-2-pyridyloxyacetic acid) is a selective post-emergent systemic herbicide used to control annual and perennial weeds (Zand et al., 2007 [25]; Abouziena et al., 2007 [26]). Once absorbed into the weeds, fluroxypyr accumulates throughout growth and degrades slowly. When fluroxypyr combines with auxin receptor sites in the plant cell, weed growth is disturbed by the deregulation of cellular growth processes (Grossmann 2010 [27]; LeClere et al., 2018 [28]). Fluroxypyr interferes with the ability of weeds to metabolize nitrogen and produce enzymes. When growth regulation is disrupted, weed growth becomes disorganized, disrupting metabolic processes, which results in weed death. Numerous studies have investigated the impact of fluroxypyr on crops, which mainly manifests in two aspects: a decrease in crop growth (Breeze 1988 [29]; Guo et al., 2018 [30]) and the inhibition of photosynthesis. Wu et al. (2009) [31] found that fluroxypyr led to the chlorosis of leaves by suppressing chlorophyll formation in rice. Guo et al. (2018) [30] demonstrated that fluroxypyr restrained the growth of millet, as reflected by a decrease in plant height. In this study, second-generation Illumina MiSeq sequencing technology was used to provide a more direct method to detect microbial populations, especially the changes in species with low abundance (Hong et al., 2015 [32]). The dynamic changes in soil microbial communities provide a basis for the preliminary study of the effects of fluroxypyr on corn growth, yields, and crop quality and provide a basis for the further discussion of the effects of pesticides on crop quality, aiming to guide the safe use of pesticides.

## 2. Experimental Procedures

### 2.1. Study Site and Fluroxypyr Treatment

This study was completed from April 2020 to February 2023. The study site was located in the experimental base of the Maize Research Institute, Shandong Academy of Agricultural Science (N36°44′, E117°22′), Shandong Province, China. This area is an important maize production region, with a mean temperature of 12.8 °C and annual rainfall of 600 mm. The soil was a cinnamon loess. Under the USDA texture classification system, the soils were classified as chromic cambisols. The initial soil pH was 7.9, and the organic matter and total N, P, and K levels were 13.65, 1.41, 0.63, and 2.41 g·kg^−1^, respectively. Soil samples were collected from a field-planted Ludan 818, which is a compact hybrid maize variant (*Zea mays* L.) cultivated by the Maize Research Institute. The fields had no history of herbicide treatment over the two previous years. 

The study plot, covering an area of 1350 m^2^, was divided into six subplots (15 × 15 m) of 225 m^2^ each. Three subplots were treated with fluroxypyr, while the remaining three served as controls, treated without fluroxypyr. A randomized block design was used, ensuring uniform and homogeneous cultivation and management conditions across all plots to minimize environmental variations that might confound the herbicide’s effects. To reduce the impact of weeds on maize growth during the sowing period, the land was plowed to a depth of 15 cm before sowing. Fluroxypyr (Starane 200 EC, Dow AgroSciences, Indianapolis, IN, USA) was applied as an aqueous spray using a hand-operated knapsack sprayer, with a spray volume of 250 L ha^−1^. The herbicide was applied at a rate of 150 g active ingredient (ai) ha^−1^ during the vegetative stages. The application followed the standard protocol provided by the manufacturer, which recommended this rate for optimal weed control during these stages of growth.

### 2.2. Soil Sampling

To ensure a consistent soil sampling distance, a stainless-steel soil auger with an inner diameter of 34 mm was used to collect soil samples from the top 0–15 cm. The maize roots were shaken vigorously to remove loose soil. The soil that remained adhered to the root hairs was sampled as the rhizosphere soil (Gangola et al., 2023 [33]). The non-rhizosphere soil was sampled 5 cm away from the maize rhizosphere. For endophyte collection, maize root fragments were treated as described by Gangola et al. (2021) [34]. The fragments were immersed in 90% ethanol for 5 min, followed by 10 min in 3% sodium hypochlorite solution and then 30 s in 75% ethanol. The fragments were rinsed with sterile water for 30 s. This procedure was repeated 5–7 times to ensure proper disinfection.

Samples were collected 7 days after fluroxypyr application at three different maize growth stages: seedling, vegetative, and flowering and fruiting. Each treatment had three replicated plots, and soil samples were collected from the rhizosphere and non-rhizosphere compartments at the Maize Research Institute, Shandong Academy of Agricultural Sciences. Rhizosphere soil: A criss-cross sampling method was used to collect rhizosphere soil. In each plot, 5–10 maize plants were randomly selected, and the soil firmly attached to the maize roots was collected by gently shaking off the loose soil. The soil samples from these plants were then thoroughly mixed and stored in a low-temperature sample box. Non-rhizosphere soil: A W-shaped sampling method was employed to collect non-rhizosphere soil. After removing the topsoil, a 3.5-cm-diameter soil sampler was used to collect soil from a 0–15 cm depth at a distance of 20 cm from the main stem of the maize. Samples were taken from 15 randomly selected points in each plot, and the collected samples were mixed well before being transported in a low-temperature storage box. A total of 15 composite samples were collected from the experimental site during June (seedling stage), August (flowering stage), and September (fruiting stage) of 2016. Each composite sample was formed by pooling soil from the respective treatment plot at each sampling time, ensuring consistency across the three growth stages. All soil samples were sieved through a 2 mm mesh and divided into two portions: one was stored at −70°C for molecular biology analysis, and the other was air-dried and finely ground for the determination of soil physical and chemical properties. No fluroxypyr was used in any of the six plots at the seedling stage, so three samples were taken. Fluroxypyr was applied in three plots at the flowering stage, while the remaining plots did not receive fluroxypyr, and six samples were taken at this stage. At the fruiting stage, six samples were collected from both the treated and control plots. Sample-specific details are provided in Appendix A. All samples were carried on ice to be preserved in the laboratory at 4 °C. The non-rhizosphere, rhizosphere, and endophyte samples treated with fluroxypyr were labeled as NR, R, and E; the control samples were labeled as NRCK, RCK, and ECK, respectively. The sample identifiers 1, 2, and 3 represent the samples that were obtained in the seeding, florescence, and fruiting stages, respectively. 

### 2.3. Analysis of Parameters of Maize

Five representative plant samples were obtained from every subplot of the fluroxypyr-treated maize (F) and non-fluroxypyr-treated maize (NF) during the fruiting stage. The 100-grain weight, ear weight, ear length, and bald tip length were determined. In addition, the morphology of the maize roots was observed.

### 2.4. DNA Extraction, Polymerase Chain Reaction (PCR) Amplification, and Illumina MiSeq Sequencing

Total DNA was extracted from samples using the Rapid Bacterial Genomic DNA Isolation Kit (Sangon Biotech, Shanghai, China). The concentration and quality of the extracted DNA were examined using ultraviolet spectrophotometry and 0.8% agarose gel electrophoresis. DNA was extracted from three maize plants per treatment group (control and fluroxypyr-treated) to ensure the representativeness of the microbial community associated with maize roots. The V4-V5 region of the 16S rRNA gene was amplified using primers 530F (GTGCCAGCMGCCGCGGTAA) and 907R (CCGTCAATTCMTTTRAGTTT). The PCR reaction system was as follows: 25 µL total volume—12.5 µL 2× Taq PCR Master Mix (Sangon Biotech, China); 1 µL of each primer (10 µM); 2 µL DNA template; 8.5 µL ddH_2_O. The amplification reactions were run using the following cycling parameters: initial denaturation at 95 °C for 2 min; 30 cycles of 95 °C for 15 s, 55 °C for 30 s, and 72 °C for 30 s; and a final extension at 72 °C for 5 min. The PCR products were checked using 2% agarose gel electrophoresis, and the target fragment was gel-purified using a Gel Recovery Kit (Axygen, Union City, CA, USA). The purified PCR products were then sequenced by Illumina MiSeq (Illumina, San Diego, CA, USA), following the manufacturer’s instructions, at Shanghai Personal Biotechnology. For sequencing, paired-end reads were generated with a read length of 250 base pairs for both forward and reverse reads. Library preparation and indexing were performed using the Illumina Nextera XT DNA Library Preparation Kit (Illumina, USA). The library was then loaded onto the Illumina MiSeq platform, and sequencing was conducted following the standard protocols provided by the manufacturer.

### 2.5. Statistical Analyses

The diversity and composition of the bacterial communities in the maize root samples were analyzed based on raw sequencing data obtained using the Illumina MiSeq platform. A detailed description of the methodology is provided in the Appendix A. Raw sequences were processed following the methods outlined by Magoč and Salzberg (2011), using paired-end sequencing. Sequencing reads were assigned to each sample based on the corresponding unique barcode. Low-quality sequences shorter than 150 bp were filtered out prior to further analysis. Taxonomic identification and classification of the operational taxonomic units (OTUs) or amplicon sequence variants (ASVs) were performed using the QIIME pipeline (Caporaso et al., 2010 [35]). The data were analyzed for diversity and abundance using the R software package (R 4.2.1) and the Mothur software package (v.1.48.0). Richness and diversity indices, including Chao1, ACE, Shannon, and Simpson, were calculated. For the principal coordinate analysis (PCoA), the data were standardized by log-transformation, ensuring comparability across samples. Rarefaction curves were generated to check for a sufficient sequencing depth before calculating alpha diversity indices. To remove potential plant organelle sequences, sequences identified as originating from maize plant DNA were filtered out based on reference databases and the sequence length. Statistical analysis was performed with SPSS V22.0, and normality tests were conducted prior to an ANOVA. The Tukey honestly significant difference (HSD) test was used to compare significant differences across groups. The heatmap and clustering analysis were achieved using the pheatmap (version 1.0.12) and ggplot2 (version 3.3.3) packages in R (version 4.0.3). The data were visualized and interpreted to explore patterns in the microbial community composition. All raw sequencing data are available in the NCBI SRA database (https://www.ncbi.nlm.nih.gov/sra accessed on 16 October 2019) (BioProject ID PRJNA577789), with 15 SRA libraries corresponding to the root endosphere of maize, derived from multiple samples per compartment.

## 3. Results

### 3.1. Similarities in the Bacterial Community in Different Portions of Maize Roots

After quality sequencing, the abundance of all samples in different operational taxonomic units (OTUs) was obtained by clustering the OTUs at the 97% similarity level. Common bacterial OTUs are presented to assess the relationships among the communities (Figure 1). The numbers of OTUs obtained were 5152 for NR (non-rhizosphere samples treated with fluroxypyr), 5266 for R (rhizosphere samples treated with fluroxypyr), and 2667 for E (endophyte samples treated with fluroxypyr). NR and R shared 4866 OTUs, NR and E shared 1806, and R and E shared 1888. 

The principal component analysis highlighted the obvious separation of the E bacterial communities. NR and R were concentrated together, indicating a more similar community structure between the two (Figure 1B). The first axis explained 70.87% of the cumulative percentage variance of the species, and 21.82% was explained by the second axis. Overall, the principal components explained 92.69% of the total variation in the different communities.

### 3.2. Effects of Fluroxypyr on the Richness and Diversity of Microbial Communities

After filtering the raw reads, we obtained 333,565 high-quality reads with an average length of 394 bp. We investigated the biodiversity of the microbial communities using the Chao1, ACE, Shannon, and Simpson diversity indices (Table 1). A decrease in biodiversity across all indicators was observed after fluroxypyr application, especially from E3CK to E3 (both the Chao1 index and the ACE index fell by 62%). In contrast, changes between NRCK and NR, as well as RCK and R, were not obvious. The rank abundance curves expressed the diversity of the samples in terms of richness and evenness. The curves were generated by collating the OTUs according to their abundance, with the width of the curve representing the richness and the flatness representing the evenness. E3, treated with fluroxypyr, showed a rapid drop compared with E3CK (Figure 2). This is consistent with the observations in Table 1. These results indicate that the use of fluroxypyr affects the richness and diversity of endophytic bacterial communities in mature maize roots. 

### 3.3. Effects of Fluroxypyr on the Ear Traits and Morphological Observations of the Roots Throughout the Maturity Stage

Spraying fluroxypyr significantly affected the ear traits; the ear weight and 100-grain weight were significantly decreased in the fluroxypyr-treated maize compared with the non-treated control (Table 2). Compared with the non-treated maize, the aerial roots were smaller, and the underground roots were shorter (Appendix A). 

### 3.4. Effects of Fluroxypyr on the Soil Microbial Community Structure

The sequencing reads recovered from samples classified at the phylum level were affiliated with 20 bacterial phyla (Figure 3). In the NR and R soils, the dominant phyla (community richness greater than 1.5%) were Proteobacteria, Actinobacteria, Acidobacteria, Chloroflexi, Planctomycetes, Crenarchaeota, Gemmatimonadetes, Bacteroidetes, and Nitrospirae, and there was no significant difference in the dominant phylum among the treatment groups. Six bacterial phyla were dominant (community richness greater than 0.5%) in the endophyte samples: Cyanobacteria, Firmicutes, Proteobacteria, Actinobacteria, Acidobacteria, and Bacteroidetes. No clear changes were observed between the fluroxypyr-treated and control soils in the NR and R soils. Many bacteria are maize root endophytes, including Proteobacteria, Firmicutes, Actinobacteria, and Acidobacteria (Niu et al., 2017 [36]).

Further analyses were conducted to determine the relative abundances of the identified bacterial genera within the communities at the genus level in all samples (Appendix A). The heatmap shows the cluster analysis of the 50 most abundant genera in the bacterial communities of all samples (Figure 4). Significant differences in the bacterial community distribution were found between maize endophytes at the florescence and fruiting stages and the other samples (NR and R soils). The dominant genera in the endophytic samples of maize roots at the fruiting stage were *Lactococcus*, *Nocardiopsis*, *Acinetobacter*, *Agrobacterium*, *Sediminibacterium*, *Geobacillus*, *Pseudomonas*, *Planctomyces*, *Rhodanobacter*, and *Streptococcus* (Figure 5). Significant alterations in the relative abundances of *Lactococcus, Acinetobacter*, and *Agrobacterium* were found between the fluroxypyr-treated and non-treated endophytic maize samples in the fruiting stage. The proportion of *Lactococcus* in E3 increased from 10.3% to 18.6% compared with E3CK, and the proportions of *Acinetobacter* and *Agrobacterium* decreased from 11.5% in E3 to 4.7% in E3CK and from 8.6% in E3 to 2.1% in E3CK, respectively. 

## 4. Discussion

The interactions between plant roots and microorganisms are important for plant performance (Niu et al., 2017 [36]). The results of this study show that the community of the R soil was similar to that of the NR one. The R soils displayed more unique OTUs than the endophyte samples. This partly explains that the colonization of maize root-associated bacteria was selective and specialized from R to endophytic samples (Hardoim et al., 2008 [37]). Many studies have shown that rhizosphere soil bacteria are the main source of endophytic bacteria in plant roots (Lamb et al., 1996 [38], Gangola et al., 2022 [9]). This colonization is related to many factors, including the temperature, humidity, pH, salt, and oxygen (Handelsman and Stabb 1996 [39]). Bulgarelli et al. (2013) [40] suggested that the establishment of endophytes can be divided into two steps. First, the root system concentrates the microorganisms with the highest affinity around the roots by secreting signal molecules. The roots will then screen specific microorganisms to colonize the plants (Llirós et al., 2014 [41]).

The analysis of the bacterial phyla in the NR and R soil samples showed that the most dominant phyla were *Proteobacteria*, *Actinobacteria*, *Acidobacteria*, and *Chloroflexi*. These phyla include taxa that are commonly detected in soils and produce various effects on plant health, including beneficial and pathogenic interactions (Mendes et al., 2013 [42]). Edwards et al. (2015) [43] used high-throughput sequencing technology to study the rhizosphere microbial community composition and found that the dominant groups of rice root-associated bacteria were *Proteobacteria* and *Acidobacteria*. *Proteobacteria* is a phylum composed of several genera, including fast-growing organisms that prefer a high-carbon-content environment to maintain their high energy requirements and growth rates (Fierer et al., 2007 [44]). In contrast, members of *Chloroflexi* have been reported to survive in oligotrophic environments and may depend on the minimal nutrients available (Fierer and Jackson 2006 [45]). *Actinobacteria*, one of the major phyla observed in this study, are organic substance decomposers (Strap 2011) [46] that may be important in maintaining microbial-mediated processes when nutrients become scarce.

The use of fluroxypyr initially reduced the richness and diversity of the bacterial community in the R soil at the florescence stage. However, this reduction was not permanent, as the diversity and richness gradually recovered prior to the fruiting stage. While there were some changes in the community composition at the phylum and genus levels in both NR and R soil samples after herbicide application, the changes were relatively minor, indicating that fluroxypyr had a limited and transient effect on the overall microbial community in the maize rhizosphere. These findings suggest that the soil microbial community, especially in the R soil, showed signs of resilience over time. The reason for this may be related to the fluroxypyr application method. During this experiment, the herbicide was sprayed onto the stems and leaves of the weeds, meaning that it did not immediately enter the soil, resulting in a low concentration in the soil. As such, its effect on the rhizosphere microbial community was small. Some studies have reported that pesticides have a minimal lasting effect on the total number of microorganisms, with the impact on biological indicators diminishing as the herbicide degrades in the soil (Fang et al., 2009 [47], Bhandari et al., 2024 [48]). However, it would be valuable to include data on the herbicide concentration in the soil to support this speculation. Regarding the observed significant effects on the maize yield, it is possible that the herbicide, despite its limited impact on the microbial community, may have influenced maize growth through other indirect mechanisms, such as weed suppression.

In this study, fluroxypyr decreased the community diversity of the endophytic bacteria in the mature maize roots. We also found that corn plants with a small number of weeds grew better than plants treated with herbicides. High-throughput sequencing allowed us to perform an in-depth study of the bacterial community composition. According to the analysis of the bacterial genera in mature maize after the application of fluroxypyr, the proportions of *Acinetobacter* and *Agrobacterium* were reduced, while the proportion of *Lactococcus* increased. 

*Agrobacterium* microorganisms secrete the phytohormone indole-3-acetic acid (IAA) (Inzé et al., 1984 [49]). IAA is the earliest auxin-like substance and has an important role in cell elongation, division, and tissue differentiation (Barazani and Friedman 1999 [50]). IAA-producing bacteria may affect plant growth in three main ways: producing IAA, which directly affects plant growth; increasing the utilization of nutrients by plants; and inhibiting the growth of plant pathogens (Kuklinsky-Sobral et al., 2004 [11]; Hayat et al., 2010 [51]; Gangola et al., 2023 [33]). The plant root is the most sensitive portion to IAA, which promotes the elongation of the main root and the formation of lateral roots. In turn, the growth of the main and lateral roots increases the production of plant root exudates and provides more nutrients for rhizospheric microorganisms (Tsavkelova et al., 2007 [52]).

*Acinetobacter* is a human pathogen that is widely present in human environments and hospitals. However, we found *Acinetobacter* among the endophytes of maize roots. Gangola et al. (2018) [53] found a large number of *Acinetobacter* strains in soil and water. These strains gather to form a unique physiological metabolic mechanism that may use various carbon sources. Sachdev et al. (2010) [54] found that *Acinetobacter* is widely present in the wheat rhizosphere and has plant-promoting functions, such as nitrogen fixation, siderophore production, and mineral dissolution. *Acinetobacter* produces a large amount of xylanase and agar hydrolase to degrade hemicellulose and structural polysaccharides (Purohit et al., 2017 [55]). These studies show that *Acinetobacter* may provide nutrients that promote plant growth, such as IAA.

*Lactococcus* is a lactic acid-fermenting bacterium. Studies have shown that EM4 lactic acid-fermenting bacteria accelerate the decomposition of organic amendments in soils and the release of their nutrients for plant growth (Higa and Kinjo 2024 [56]). In this study, the proportion of *Lactococcus* in E3 increased from 10.3% to 18.6% compared with E3CK. Our results suggest that the increased amount of *Lactococcus* in the fluroxypyr endophytic fruiting stage may contribute to maize growth. However, the decreased maize growth (decreased ear length, root size, and 100-grain weight) suggests that increased *Lactococcus* flora cannot offset the negative effects of fluroxypyr on maize.

Endophytic bacteria may promote plant growth, increase the plant height, and enhance the growth potential of plants (Gangola et al., 2023) [33]. Li (2020) [57] found that *Acinetobacter* ZJU-1 chitin fermentation liquid could promote the growth of cucumber and tomato. The study of Das and Sarkar (2018) [58] showed that *Acinetobacter lwoffii* promoted the growth of mung beans by inhibiting the absorption of arsenic. Thus, we hypothesize that fluroxypyr has an effect on the ear and root characteristics of mature maize, mainly due to the decrease in *Acinetobacter* and *Agrobacterium* in endophytes. We recognize that this correlation does not necessarily imply causation. It is possible that both the changes in the microbiome composition and the observed morphological changes are independent consequences of fluroxypyr’s non-target effects, particularly due to its synthetic auxin activity. 

## 5. Conclusions

In summary, our research demonstrated that fluroxypyr inhibited maize growth by disturbing the endophytic bacterial community structure and decreasing the amounts of *Acinetobacter* and *Agrobacterium* in endophytes. These results provide data for future studies on maize production and may help to improve the understanding of the relationships between endophytes and maize growth. In the future, we will explore in situ remediation technologies with these bacteria and crop-safe production in fluroxypyr-contaminated soil under field conditions. Our results suggest that the ecotoxicology of existing and emerging hazardous materials and the connection between the bacterial community structure and crop growth are of great importance.

## Figures and Tables

**Figure 1 microorganisms-13-00728-f001:**
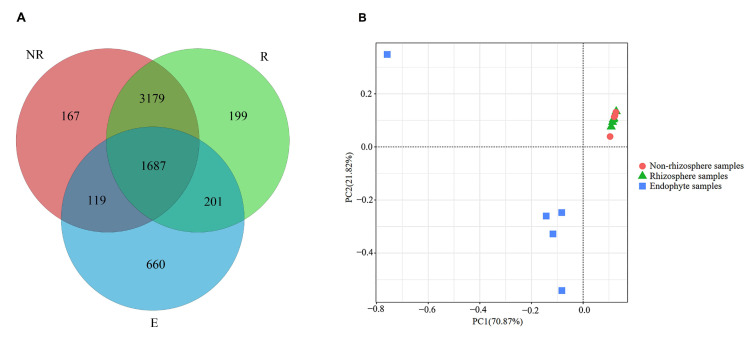
Venn diagram depicting the number of bacterial OTUs unique and shared between the samples. NR, R, and E are non-rhizospheric soil, rhizospheric soil, and endophyte samples, respectively (**A**). Principal coordinate analysis (PCoA) based on the genus distribution of the bacteria in NR, R, and E samples (**B**). The shape of the points represents the different sample types (NR, R, E), and the colors indicate the clustering of samples based on their bacterial composition. Each point represents a sample, and the distance between points reflects the dissimilarity in the microbial community composition.

**Figure 2 microorganisms-13-00728-f002:**
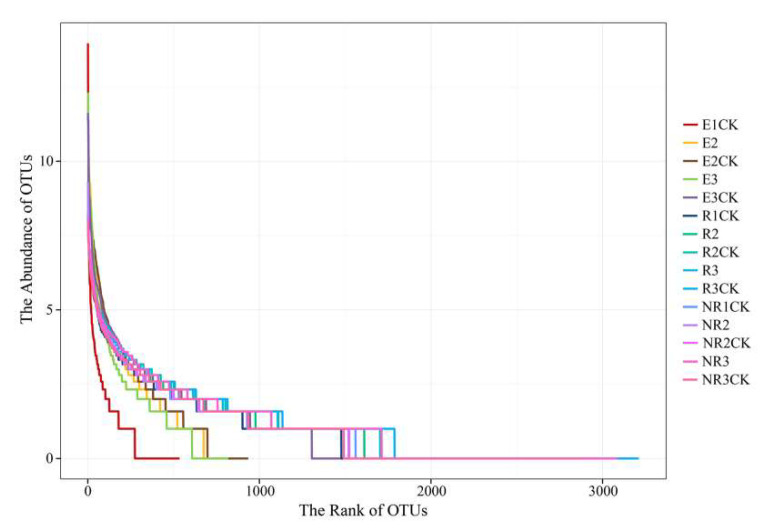
Rank abundance curves of the bacterial communities. The width of the curve represents the richness and its flatness represents the evenness. NR, R, and E are non-rhizospheric, rhizospheric, and endophyte samples treated with fluroxypyr, while NRCK, RCK, and ECK, respectively, are the control samples. The sample identifiers 1, 2, and 3 represent the samples obtained in the seeding, florescence, and fruiting stages, respectively.

**Figure 3 microorganisms-13-00728-f003:**
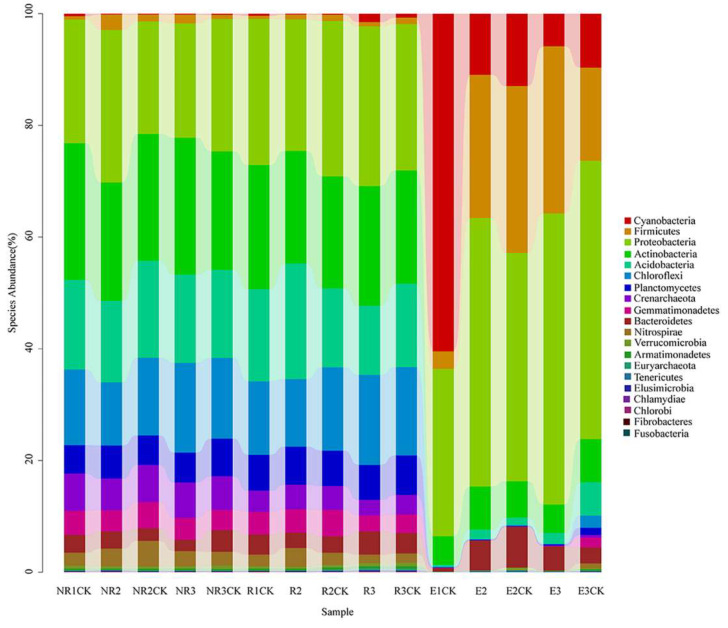
Phylum-level taxonomic composition of the bacterial community. NR, R, and E are non-rhizosphere, rhizosphere, and endophyte samples treated with fluroxypyr, while the control samples are NRCK, RCK, and ECK, respectively. The sample identifiers 1, 2, and 3 represent the samples obtained in the seeding, florescence, and fruiting stages, respectively.

**Figure 4 microorganisms-13-00728-f004:**
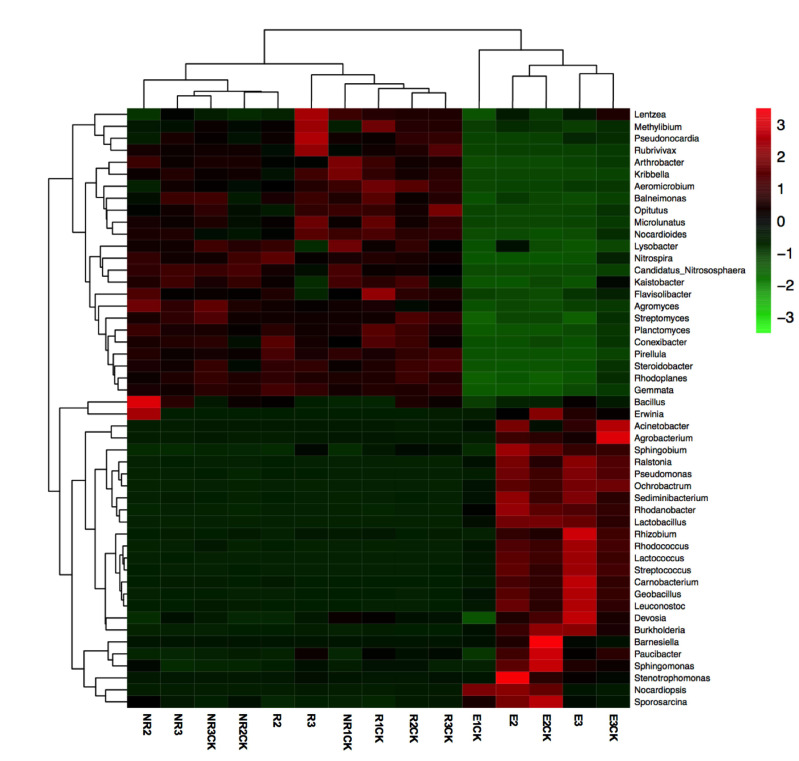
Heatmap showing the cluster analysis of the 50 most abundant genera in the bacterial communities. NR, R, and E are non-rhizospheric, rhizospheric, and endophyte samples treated with fluroxypyr, while the control samples are NRCK, RCK, and ECK, respectively. The sample identifiers 1, 2, and 3 represent the samples obtained in the seeding, florescence, and fruiting stages, respectively.

**Figure 5 microorganisms-13-00728-f005:**
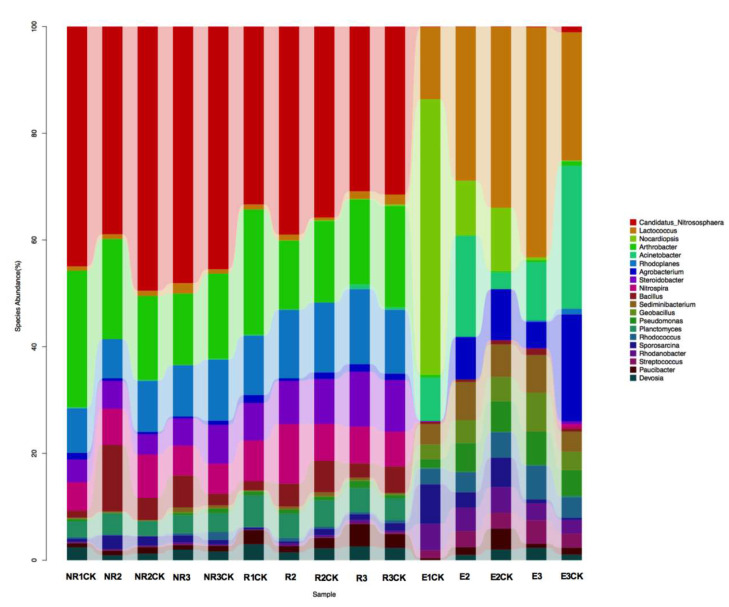
The influence of fluroxypyr on the top 20 genera identified through random forest classification. NR, R, and E are non-rhizospheric, rhizospheric, and endophyte samples treated with fluroxypyr, while the control samples are NRCK, RCK, and ECK, respectively. The sample identifiers 1, 2, and 3 represent the samples obtained in the seeding, florescence, and fruiting stages, respectively.

**Table 1 microorganisms-13-00728-t001:** Diversity indices of the soil microbial communities based on the 16S rRNA genes analyzed via Illumina MiSeq sequencing. The non-rhizosphere, rhizosphere, and endophyte samples that were treated with fluroxypyr are labeled as NR, R, and E, while the control samples are labeled as NRCK, RCK, and ECK, respectively. The sample identifiers 1, 2, and 3 represent the samples obtained during the seedling, flowering, and fruiting stages, respectively.

Sample		Chao1	ACE	Simpson	Shannon
Non-rhizosphere	NR1CK	4634	5043	0.9957	9.83
	NR2	4398	4694	0.9947	9.70
	NR2CK	4395	4717	0.9966	9.86
	NR3	4548	4835	0.9967	9.93
	NR3CK	4570	4908	0.9971	10.01
Rhizosphere	R1CK	4395	4731	0.9967	9.92
	R2	4550	4917	0.9967	9.88
	R2CK	5061	5269	0.9971	10.04
	R3	4641	4854	0.9970	10.12
	R3CK	4760	5041	0.9972	10.13
Endophyte	E1CK	888	935	0.6033	2.47
	E2	1055	1069	0.9576	6.11
	E2CK	1134	1128	0.9645	6.66
	E3	969	972	0.9292	5.57
	E3CK	2576	2593	0.9620	7.10

**Table 2 microorganisms-13-00728-t002:** Effects of fluroxypyr on the ear characteristics of maize. F represents the fluroxypyr-treated maize; NF represents the non-fluroxypyr-treated maize. The results of the ANOVA are shown at the bottom of the table. * and ** represent significant differences between fluroxypyr-treated and non-fluroxypyr-treated maize at the 0.05 and 0.01 levels, respectively.

Treatment	Ear Weight (kg)	Ear Length (cm)	Bald Tip Length (cm)	100-Grain Weight (g)
F	0.71 ± 0.02	19.63 ± 0.63	1.87 ± 0.31	39.44 ± 0.37
NF	0.75 ± 0.01	20.15 ± 0.32	1.42 ± 0.48	41.02 ± 0.30
Source of variation Treatment	0.035 *	0.275	0.248	0.005 **

## Data Availability

The datasets generated and/or analyzed during the current study are available in the NCBI SRA database (https://www.ncbi.nlm.nih.gov/bioproject/PRJNA577789 accessed on 16 October 2019) repository [PERSISTENT WEB LINK TO DATASETS].

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
