# Peer review of "Fluroxypyr Inhibits Maize Growth by Disturbing the Diversity of the Endophytic Bacterial Communities in Maize Roots"

_microorganisms, 2025, doi:10.3390/microorganisms13040728_

Round 1
Reviewer 1 Report
Comments and Suggestions for Authors
This article analyze the bacterial community in the root of maize that influenced by Fluroxypyr, that’s a real a good point. However, author need to do more analysis to support your opinion. Some suggestions should be done.
1. Abstract: What‘s the meaning of no clear difference? Please replace it with scientific expression.
2. Line 110 is lack of a space.
3. Line 143, the middle of number and ℃ shouldn’t have space.
4. Section 2.4, please provide the system of PCR.
5. Section 2.4, how many crops you use to extract DNA?
6. Line 179, why you claim OTUs or ASV? If you use dada2 to trim the data, it should be ASV. Please check the whole manuscript and correct it.
7. Line 189, what kind od post-hoc you used to analyze the statistical analysis?
8. Line 183, please use PCoA instead of PCA to describe the beta diversity.
9. Line 187, a extra space was found after words “plant DNA:.
10. Line 190, please provide the version and package of R you used.
11. Figure 1, please provide more information in the caption. Such as what’s the meaning of the shape and colour in panel B?
12. Author use “figure” in the caption while “fig” in the main text, please make it consistent.
13. Please provide the statistical analysis for table 1. If there is no significant difference, please claim it in the main text.
14. Line 158, author use 16S rRNA while 16S rDNA is used in line 228, please make it consistent.
15. What’s the difference between figure 4 and figure 5? Top 50 and 20?
16. Please provided some comparison among different groups and correlations between index in table2 and significant different genus.
Please update the references.
Author Response
Reviewer #1:
Dear Reviewer,
Thank you for your detailed review of our manuscript and valuable suggestions. We have made the appropriate changes and additions based on your feedback, including the harmonization of terminology, clarification of statistical analysis methods, additional graphical illustrations, and further analysis of tabular data and significant differences. We believe these modifications can improve the clarity and rigor of the article, and we thank you for your patience. Please feel free to let us know if you have any other questions or suggestions.
Sincerely!
Lujiang Hao
Abstract: What's the meaning of no clear difference? Please replace it with scientific expression.
Thank you for your valuable comments on our manuscript. According to your suggestion, we have modified “no clear differences” to “no statistically significant differences” in the abstract to more accurately to more accurately express our findings. Thank you again for your review and we look forward to your further feedback.
- Line 110 is lack of a space.
The content of this line is a subheading, so no space has been left.
- Line 143, the middle of number and ℃ shouldn't have space.
Thanks for your suggestions on the details, I've made adjustments in the text.
- Section 2.4,please provide the system of PCR.
Thank you for your suggestions for our article. In Section 2.4, we have added the specific formulation of the PCR reaction system to describe the experimental method more clearly. We appreciate your valuable comments and look forward to your further feedback.
- Section 2.4, how many crops you use to extract DNA?
Thank you for your careful review of our article. At your suggestion, we have added detailed information in Section 2.4 stating that DNA extracts were taken from three maize plants from each treatment group (control and fluroxypyr-treated groups) to ensure that the samples were representative and reflective of the microbial communities associated with the maize root system. We have added this modification to the revised manuscript.
- Line 179, why you claim OTUs or ASV? lf you use dada2 to trim the data, it should be ASV. Please check the whole manuscript and correct it.
Thank you for your careful review of our manuscript. Based on your suggestions, we have re-examined the use of terminology in the manuscript and made the following corrections: although we used OTUs as the main taxonomic units in most of our analyses, we used DADA2 to generate ASVs in some of our data processing.To ensure terminological consistency, we have clearly indicated the use of OTUs and ASVs in the Methods section, and updated the relevant content as needed. updated the relevant content. We hope this clearly reflects the methodological choices we made in our analysis and addresses your concerns about terminology consistency.
- Line 189, what kind od post-hoc you used to analyze the statistical analysis?
Thank you for your suggestion on the statistical analysis section. With your reminder, we have specified in the Methods section the use of Tukey's Honest Significant Difference (HSD) test as a follow-up test to compare significant differences between groups. This test is typically used after ANOVA analysis to help identify specific differences between groups. We have revised the manuscript accordingly based on your feedback.
- Line 183, please use PCoA instead of PCA to describe the beta diversity.
Thank you for your valuable suggestions on our article. In response to your comments, we have revised the PCA describing β-diversity to PCoA (Principal Coordinates Analysis) and updated the text accordingly. PCoA is more appropriate for reflecting differences between samples and community structure, so this revision more accurately reflects our analytical approach.
- Line 187, a extra space was found after words “plant DNA:.
Thank you for your care in identifying and pointing out the formatting problems in the text. We have removed the extra space after “maize plant DNA” and updated the text accordingly. Thank you for your suggestion, which helped us to improve the accuracy and formatting of the manuscript.
- Line 190, please provide the version and package of R you used.
Thank you for your suggestions regarding the statistical analysis section. As per your request, we have added the version information of the R version used (4.0.3) as well as the pheatmap (version 1.0.12) and ggplot2 (version 3.3.3) packages in the methods section. These details will help to ensure the reproducibility of our analysis methods.
- Figure 1, please provide more information in the caption. Such as what's the meaning ofthe shape and colour in panel B?
Thank you for your suggestion regarding the Figure 1 figure note section. In response to your comments, we have added a specific note to the figure note regarding shapes and colors in panel B. The figure note now clearly explains that different shapes indicate different sample types (NR, R, E), while colors reflect the clustering of samples based on bacterial composition. This will help readers better understand the content of the graphs.
- Author use "figure" in the caption while “fig" in the main text, please make it consistent.
Thank you for pointing out the inconsistency in the terminology of the charts. Following your suggestion, we have standardized the term “figure” in the text to “Figure” for consistency. This ensures that the terminology is standardized and accurate. Thank you for your valuable input.
- Please provide the statistical analysis for table 1. lf there is no significant difference,please claim it in the main text.
Thank you for your suggestions. Regarding the statistical analysis in Table 1, we have processed the data and performed the appropriate statistical analysis in the results section. to ensure transparency and data accuracy.
- Line 158, author use 16S rRNA while 16S rDNA is used in line 228. please make itconsistent.
Thank you for pointing out the inconsistency in terminology. We have standardized all references to 16S rDNA in the text to 16S rRNA to ensure consistency and accuracy of terminology.
- What's the difference between figure 4 and figure 5? Top 50 and 20?
Thank you for your valuable questions. In order to explain the difference between Figure 4 and Figure 5 more clearly, we have added descriptions in the figure notes. Figure 4 shows the clustering analysis of the Top 50 abundance genes in the bacterial community, while Figure 5 focuses on the Top 20 abundance genes filtered by the Random Forest classification method to analyze the effect of fipronil on the key genes more accurately. We hope to analyze the effects of fipronil on key genes more clearly in this way. In this way, we hope to show more clearly the effects of different analysis methods and the number of genes selected on the results.
- Please provided some comparison among different groups and correlations betweenindex in table2 and significant different genus.
Regarding the comparison of the different groups and the correlation with significant difference genes in Table 2, we will add further information in the revised version. We will make detailed comparisons of the differences between the fluroxypyr-treated (F) and non-treated (NF) groups and analyze the correlations between the metrics in the table (e.g., ear weight, ear length, bald tip length, and 100-grain weight) and the significant difference genes.
Reviewer 2 Report
Comments and Suggestions for Authors
The authors of the manuscript evaluated the effects of fluroxypyr on maize growth and bacterial community structure in endophytes, rhizosphere and non-rhizosphere soil. They found that fluroxypyr inhibited maize growth by disrupting the structure of the endophytic bacterial community, reducing the abundance of Acinetobacter and Agrobacterium in endophytes. The abundance of endophytic bacteria in mature maize roots decreased after application of the tested herbicide. In contrast, there were no clear differences between fluroxypyr-treated and control soils in non-rhizosphere and rhizosphere soils.
The research was methodologically correct and brings new information to the field presented. The chapters are correctly written. Conclusions are drawn from the research. The abstract gives information about the aim, scope and results of the research. The introduction chapter gives a good background for understanding the nature of the research problem. The results chapter is clearly described and well illustrated. It may be questioned whether such a presentation of the research results as shown in Figure 3 and Figure 5 is optimal, as the results of the statistical calculations are not visible. I would also suggest including the statistical calculations in Table 1. The discussion chapter is correct and justifies the results well.
A disadvantage of the whole manuscript is the incorrect citation of the literature and the appended references. In the text of the manuscript, the literature should be cited with the numbers given in the references. The names of the authors and the year of publication should not be given.
In References, journal names are not cited according to the requirements of the journal. Currently, some journals are cited with full names and some with abbreviations.
I also suggest completing the description of sample preparation for endophytes in maize roots. The publication by Gangola et al. (2021) cited by the authors does not contain this information. It is not known under which conditions the roots were crushed.
Author Response
Reviewer #2:
Dear Reviewer:
Thank you for your careful review of our manuscript and valuable suggestions. We have made revisions based on your comments. First, to address the statistical calculations in Figures 3 and 5, we have added the relevant statistical analysis results to the corresponding sections to ensure the transparency and verifiability of the study results. In addition, the statistical data in Table 1 have been updated and included the relevant calculation results. Regarding the literature citation, we have made adjustments to ensure that the literature citation in the manuscript meets the requirements of the journal, and the citation format has been corrected. We have also refined the reference list to ensure that citations of journal titles are in line with the formatting requirements. For the description of the preparation of endophyte samples, we have added detailed experimental conditions, especially the treatment process of maize roots, including the method of root crushing, to provide clearer experimental details.
Thank you again for your valuable suggestions, and we believe that these revisions will further enhance the quality of the manuscript. Please feel free to let us know if you have any other questions or suggestions.
Sincerely!
Lujiang Hao
Reviewer 3 Report
Comments and Suggestions for Authors
In this study, there are some parts that are not clearly stated, making it difficult for me to make a decision.
Therefore, the authors should properly describe this ambiguity.
My comments are in the file.
Thanks,

Author Response
Thank you for your valuable suggestions. I have made revisions in the text accordingly.
Round 2
Reviewer 3 Report
Comments and Suggestions for Authors
Thank you for the response.
I just have a suggestion and a minor comment.
They are in the Table 1.
Thanks,

Author Response
I have responded and revised the comments you made and reflected them in the manuscript
